# Characterization of the Bacterial Communities Inhabiting Tropical Propolis of Puerto Rico

**DOI:** 10.3390/microorganisms11051130

**Published:** 2023-04-26

**Authors:** Ana E. Pérez Matos, Giovanni Bacci, Luigimaria Borruso, Maria Landolfi, Dominique Petrocchi, Sonia Renzi, Brunella Perito

**Affiliations:** 1Biotechnology and Agrobiotechnology Research and Learning Center, Department of Natural Sciences, Pontifical Catholic University of Puerto Rico, Ponce 00717, Puerto Rico; 2Department of Biology, University of Florence, Via Madonna del Piano 6, Sesto Fiorentino, 50019 Florence, Italy; 3Faculty of Science and Technology, Free University of Bozen/Bolzano, 39100 Bolzano, Italy; 4Scientific Laboratory of Opificio delle Pietre Dure, Viale F. Strozzi 1, 50129 Firenze, Italy

**Keywords:** tropical propolis, bacterial community, antimicrobial activity, honeybees, metataxonomy

## Abstract

Propolis is a resinous material produced by honeybees from different plant sources and used in the hive as a building material and to protect the colony from parasites and pathogens. Despite its antimicrobial properties, recent studies showed that propolis hosts diverse microbial strains, some with great antimicrobial potential. In this study, the first description of the bacterial community of propolis produced by the gentle Africanized honeybee was reported. Propolis was sampled from hives of two different geographic areas of Puerto Rico (PR, USA), and the associated microbiota investigated by both cultivation and metataxonomic approaches. Metabarcoding analysis showed appreciable bacterial diversity in both areas and statistically significant dissimilarity in the taxa composition of the two areas, probably due to the different climatic conditions. Both metabarcoding and cultivation data revealed the presence of taxa already detected in other hive components and compatible with the bee’s foraging environment. Isolated bacteria and propolis extracts showed antimicrobial activity against Gram-positive and Gram-negative bacterial tester strains. These results support the hypothesis that the propolis microbiota could contribute to propolis’ antimicrobial properties.

## 1. Introduction

Propolis is a waxy and resinous substance produced from leaves, buds, balsams, and other plant materials. Honeybees break these materials with their mandibles and process them by using enzymes, salivary secretions, wax, and pollen. The cementing bees make the resulting resin smooth, with their mandibles forming the mature propolis and attaching it along the hive walls by filling cracks or gaps and forming a protective envelope [1]. The term propolis derives from two ancient Greek words, “pro” and “polis”, and it means literally “at the entrance of the city”; in fact, the main role of propolis is the defense of the hive, forming a protective barrier against environmental agents and external invaders (lizards, insects, and mice) [1], as well as pathogens and parasites, thanks to its antibacterial, antifungal, antiviral, and antiparasitic activities [2,3].

The chemical composition of propolis is highly complex and varies depending on the geographical area of production, seasonality, time of collection, altitude [4], and botanical source of collection [2]. Raw propolis typically comprises 50% plant resins, 30% waxes, 10% essential and aromatic oils, 5% pollens, and 5% other organic substances [5]. The most abundant constituents of most propolis samples are polyphenols, mainly phenolic acids and flavonoids [6]. The rich chemical composition and the abundance of bio-active molecules confer a wide spectrum of biological activities to propolis, such as antimicrobial, antioxidant, anti-inflammatory, immunomodulatory, antiparasitic, and antiproliferative [7]. These activities are exploited in the beehive and in folk medicine and are mainly investigated in propolis extracts [8]. One of the most studied aspects is its antimicrobial activity, which contributes to hygiene in the beehive [4]. Indeed, propolis and its extracts show strong and broad-spectrum antimicrobial activity in several in vitro studies. They are active against a wide range of bacterial species, such as *Pseudomonas aeruginosa* and *Salmonella typhi*, fungal species belonging to the *Candida* genus [5], and virus species, especially *Herpes* viruses and *Influenza* viruses [9]. The observed antimicrobial activity appears to be due to the mixed action of different chemical components, especially phenols, and it also depends on the properties of the plant sources used; the chemical complexity of the propolis, resulting from the mix of resins from different plant sources, makes the development of resistance difficult for parasites and pathogens [10,11].

In addition to the protection against pathogens, the antimicrobial properties of propolis can also contribute to keeping the colony healthy by maintaining a stable or homeostatic microbial community. It has been demonstrated that propolis can reduce the overall taxonomic diversity of the honeybee microbiota and can limit changes in the overall microbial community in the hive [12]. Furthermore, a decrease in the hive’s microbial load is associated with a decrease in the bee’s immune response investment, leading to a lifespan increase [11].

Bees and beehives offer several micro-niches that could be colonized by rich microbial communities. Indeed, the hive and bee microbiota are now considered a component of the hive itself as they can contribute to the health of the hive and the bee community. In the literature, much information is available about the composition of the bee’s gut microbiome and its role in health [13,14,15]. Fewer studies investigate the microbial communities associated with the other hive components and bee products [16,17,18,19], and the functions that they serve in bees’ health and in maintaining the beehive. The microbial community found in honey, for example, is mainly constituted by fermentative bacteria and yeasts involved in product preservation and in the process of production of the honey itself, and recent studies suggest that the honey community is dependent on the variety of floral nectars used by the bees [19].

Grubbs et al. (2015) [20] investigated the possible presence of microbial communities in different beehive components using membrane lipid biomarkers. Although propolis has been considered relatively aseptic due to its antimicrobial activities, it presented the most diverse community profile and a higher number of unique lipids. The detected microbial membrane lipids in propolis suggested the presence of viable microorganisms, which could play a role in propolis’ properties. Garcia-Mazcorro et al. (2019) [21] demonstrated the presence of different bacterial and fungal genera in Mexican propolis samples by targeted metataxonomic analysis. According to these studies, Casalone et al. (2020) [4] investigated the diversity of the microbial community of Italian propolis of *Apis mellifera* by both 16S rRNA targeted metagenomics analysis and cultivation. They demonstrated the presence of cultivable bacterial cells and a variety of microbial strains belonging to taxa already described in other hive components. The isolated bacteria showed antimicrobial activity against Gram-negative and Gram-positive bacteria and entomopathogenic fungi, with different inhibition spectra. Moreover, metataxonomic analysis revealed the presence of bacteria and fungi with a great capacity to outcompete potentially harmful microorganisms. The authors concluded that the characterized microbiota could contribute to propolis’ overall antimicrobial properties and its “disinfectant” role within the hive. The presence of fungi with antibacterial activity was also demonstrated in a study on fungal communities associated with Tunisian propolis [22]. More recently, Ersoy Omeroglu et al. (2023) [23] identified, by metabarcoding analysis, the dominant bacterial communities of pre-enriched propolis samples of *Apis mellifera* collected in Anatolia.

In the archipelago of Puerto Rico, the most widespread honeybee species is a race obtained in South America in 1956, known as the Africanized honeybee [24]. This race was obtained by hybridization of the African *Apis mellifera scutellata* and several European *Apis mellifera* subspecies (such as *A. m. iberica, A. m. ligustica, A. m. caucasia* [25]). The Puerto Rican race is Africanized by maternal descent, but it presents a high frequency of European alleles [26]; their morphology, queen population rate, high productivity, and resistance to parasites (in particular to *Varroa destructor* mites) are specific to the Africanized race. On the other hand, they have reduced defensive behavior, a feature typical of the Puerto Rican race; for this reason, these bees are known as gentle Africanized honeybees [27]. Given their resistance to parasites, it would be interesting to investigate if and how propolis can contribute to this feature.

The present study aimed to characterize the bacterial community of tropical propolis produced by the gentle Africanized honeybee of Puerto Rico through cultivation and targeted metagenomics. The antimicrobial potential of the isolated bacteria and propolis extracts was also evaluated.

## 2. Materials and Methods

### 2.1. Study Areas and Hive Description and Sampling

Propolis was sampled from 14 artificial hives of two apiaries built by local beekeepers in two different geographic and climatic areas of Puerto Rico: the Arroyo area in the southeast and the Yauco area in the southwest of the island (Figure 1). The Arroyo area is in the moist forest, where rains are abundant, and evergreen vegetation with very tall trees is present. Yauco area is in the dry forest, where rains are scarce, the vegetation covers the ground, and fire is very common. All sampling procedures took place in October 2018. In Arroyo’s apiary, propolis was collected from six gentle Africanized honeybee hives in a small forest area a few meters from the other one. All the beehives were newly installed after Hurricane Maria (20 September 2017), and only one (Lucia) survived the storm. After the hurricane, the bees left the beehives because they were stressed, and the lack of flowers led them to emigrate. Cuquita and Duke were installed in November 2017, the others in April–May 2018. In the Yauco apiary, propolis was collected from eight beehives of gentle Africanized honeybees located a few meters from one another in a prairie field. Moreover, in this case, the hives were young at the sampling time because they were installed after Hurricane Maria. Artificial beehives are usually made of wood and present different levels: a bottom board with an entrance for the bees; a brood box, where queens lay eggs; a honey super, where there are special frames where honey is stored; and an inner cover, with a central opening, separating the honey super and outer cover, which presents a metal foil to protect the hive from external agents. The propolis samples were taken from the edges of the hive structures, under the outer and the inner cover, and from the edges of frames (Figure 1C). Sampling was done by scraping the surface with a sterilized metal spatula; then, samples (each sample approximately 2 g) were collected in sterile tubes and stored at 4 °C in a portable cooler box until arrival at the laboratory. Propolis samples and their processing are listed in Table 1.

### 2.2. Cultivation of Bacteria from Propolis

Six propolis samples from the following beehives were processed to cultivate bacteria: Lucia and Florencia from the Arroyo area, and #1, #2, #8, and #13 from the Yauco area. Approximately 1 g of propolis was taken from each sample and shredded manually with a sterile metal spatula, alternating with resting phases at 4 °C for a few minutes. The dust and small fragments obtained were suspended in 3 mL of phosphate-buffered saline (PBS; 8 g/L NaCl, 0.2 g/L KCl, 1.44 g/L Na_2_HPO_4_, 0.24 g/L KH_2_PO_4_, pH 7.4) and vortexed for few minutes. After a resting phase to settle the suspension, the supernatant was removed and placed in a new sterile tube. Then, 0.1 mL of the undiluted and PBS-diluted (10-1, 10-2, 10-3, and 10-4) propolis suspensions was plated on LB agar medium. To verify that the cultivated microorganisms could derive from resistant forms, the Florencia, #2, and #8 suspensions were also treated at 100 °C for 10 min and plated on LB agar medium as described above. All the plates were incubated at 30 °C for a week. Bacterial viable titer was calculated as the mean value of the number of colony forming units (CFUs) per gram of propolis. The colonies grown from propolis suspensions were observed by the naked eye and under a stereomicroscope (Leica ES2, Wetzlar, Germany) to evaluate their morphology and identify different morphotypes. For the morphological classification, nomenclature with alphabetical letters (A, B, C, D, E, F, G, H, L) was chosen for each hive. The same letter does not identify the same morphotypes in different hives. Each morphotype was isolated on LB agar and stored as glycerol stocks at −80 °C.

### 2.3. DNA Extraction from Isolated Bacteria, 16S rRNA Amplification, and Sequencing

Genomic DNA was extracted from bacterial isolates with the Quick-DNA Fungal/Bacterial Miniprep Kit (Zymo Research, Irvine, CA, USA), following the manufacturer’s instructions, and by thermal lysis, as described in Casalone et al. (2020) [4]. The whole 16S rRNA gene was amplified by PCR using GoTaq G2 Hot Start Green Master Mix (Promega, Madison, WI, USA) and primers 8F (5′-AGAGTTTGATCCTGGCTCAG-3′) and 1492R (5′-GGTTACCTTGTTACGACTT-3′), using the following program: an initial denaturation step at 95 °C for 2 min; 34 cycles at 95 °C for 30 s, 55 °C for 30 s, 72 °C for 2 min; and the last step at 72 °C for 5 min. The amplified DNA was then purified using a QIAquick PCR Purification Kit, and its concentration was assessed by NanoDrop (UV5Nano; Mettler Toledo, Columbus, OH, USA). Purified DNA was sent to MCLAB in San Francisco, CA, USA, for sequencing. The resulting sequences were analyzed by BLAST in the National Center of Biotechnology Information database (NCBI, http://www.ncbi.nlm.njih.gov/ (accessed on 10 January 2023)) and deposited in the NCBI database under the accession numbers from OQ288109 to OQ288124.

### 2.4. Antimicrobial Activity of Propolis Extract

Six samples from Arroyo’s apiary (Lucia, Akira, Cuquita, Diabla, Duke, Florencia) and six samples from Yauco’s apiary (#1, #2, #4, #8, #11, #13) were used to test the antimicrobial activity of propolis extracts. A solution 1:10 (*w*/*v*) (0.1 g propolis in 1 mL ethanol) of each propolis sample was created in 70% ethanol, briefly mixed by vortexing, and left for a week at room temperature. Every two days, the suspensions were shaken for a few hours to encourage propolis extraction. An aqueous extract was prepared only for the #4 sample, following the same protocol. After one week, all the extracts were centrifuged and filtrated (Corning filters with porosity 0.2 µm). Propolis extracts were tested against the following microbial tester strains: the Gram-positive bacteria *Bacillus subtilis* ATCC 6633, *Staphylococcus aureus* ATCC 25923, *Enterococcus faecalis* ATCC 29212; the Gram-negative bacteria *Escherichia coli* ATCC 35218, *Acinetobacter baumannii* ATCC 19606. The bacterial strains were kept as stock cultures in glycerol at −20 °C, and each isolate was streaked on LB agar medium and incubated at 37 °C overnight. An isolated colony was taken from each strain’s plate by a sterile loop, inoculated in LB medium, and incubated at 37 °C in a shaking incubator at 120 rpm for a night. Afterwards, every suspension was diluted 1:10 and 1:100, and 0.1 mL of the bacterial suspensions with a 0.010 ≤ A600 ≤ 0.060 was plated on LB agar medium. In each plate, 5 µL of each propolis extract was spotted. As a control, 5 µL of the following solutions was also spotted: ampicillin (1 mg/mL), tetracycline (0.1 mg/mL, 10 µg/mL), 70% ethanol, and distilled water. The Lucia alcoholic extract, #4 alcoholic extract, and 70% ethanol were also tested as 1:2, 1:4, and 1:8 dilutions. Plates were incubated at 37 °C for three days. Positive results were evaluated as the appearance of an inhibition halo around the spots.

### 2.5. Antimicrobial Activity of Bacterial Isolates

The production of antimicrobials by bacteria isolated from propolis was tested by a colony picking test and a cross-streaking test against the following tester strains: the Gram-negative bacteria *Serratia marcescens* ATCC 14041, *Escherichia coli* JM109, *Klebsiella pneumoniae* NCTC 13442; the Gram-positive bacteria *Kocuria rizophila* ATCC 9341, *Staphylococcus aureus* ATCC 12600, *Bacillus cereus* ATCC 13061. The bacterial strains isolated from propolis were #13A, #13D, #13F, #13G, LUCIA A, LUCIA C, LUCIA H, LUCIA L, FLORENCIA B, FLORENCIA C, FLORENCIA F, #2C. For the colony picking test, the procedure described by Hettiarachchi et al. (2017) [28] was used. The tester strains were inoculated in LB broth and incubated at 30 °C overnight. The next day, each liquid culture was standardized, comparing the turbidity to the McFarland Standard 0.5 following the standard procedure, obtaining an approximate bacterial suspension of 1.5 × 108 cells per mL, and 100 µL of the resulting suspension was plated in duplicate for each tester strain on LB. Then, single colonies were picked up from pure cultures of each propolis strain and placed on the plates. Plates were incubated at 30 °C for 24 h. If a tester strain was inhibited by a propolis bacterial strain, an inhibition halo would be visible around the inhibiting colony. For the cross-streaking test, the method described by Casalone et al. (2020) [4] was used. Bacteria isolated from propolis were cultivated in LB agar and incubated at 30 °C. Once grown, each strain was plated on half Petri dishes in quadruplicate, and the dishes were divided into two sets: the first duplicate set was incubated at 30 °C for 4 days, while the second set was incubated for 10 days. After incubation, the tester strains were plated in the other half of the Petri dish in streaks perpendicular to those of the propolis bacteria, and then incubated again at 30 °C for 1 day. As a control, the tester strains were plated in duplicate Petri dishes without propolis bacteria. The antimicrobial activity was detected by observing growth inhibition along the streaks.

### 2.6. Total DNA Extraction from Propolis and Library Preparation

Total DNA was extracted using a Biofilm DNA Isolation Kit (Norgen Biotek Corp., Thorold, ON, Canada) or PowerBiofilm DNA Isolation Kit (MOBIO Laboratories, Inc., Carlsbad, CA USA), with some modifications [4], from approximately 50 mg of the following propolis samples: Lucia, Florencia, Diabla, Duke, Cuquita, #1, #2, #6, #11, and #16. The quality of the extracted total gDNA was checked with 1.5% *w/v* agarose gel electrophoresis. The dsDNA concentration (ng/μL) was measured using a Qubit™ 4 Fluorometer and the 1× dsDNA High Sensitivity Assay Kit (Thermo Fisher Scientific, Milan, Italy). Then, 16S rRNA gene amplicon libraries were prepared according to the Illumina 16S Metagenomic Sequencing Library Preparation protocol (Illumina, 2013), with some modifications. PCR amplification of the V3–V4 hypervariable regions (~460 bp) was performed using KAPA HiFi 2× Hot Start Ready Mix (Roche Diagnostics SpA, Monza, Italy) with the following primers, 341F (5′-TCGTCGGCAGCGTCAGATGTGTATAAGAGACAGCCTACGGGNGGCWGCA-3) and 805R (5′-GTCTCGTGGGCTCGGAGATGTGTATAAGAGACAGGACTACNVGGGTWTCTAATCC-3′) [29], with overhang Illumina adapters (bold). The thermal conditions were 95 °C (3 min), then 35 cycles of 95 °C (30 s), 55 °C (30 s), 72 °C (30 s), then a final extension of 72 °C (5 min). Libraries were purified using KAPA Pure beads (Roche Diagnostics SpA, Monza, Italy), and then dual indices from the Illumina Nextera XT Index Kit v2 (Illumina, San Diego, CA, USA) were added to the target amplicons in a second PCR step using Kapa Hot Start HiFi 2× Ready Mix DNA polymerase. The thermal conditions were 95 °C (3 min), then 8 cycles of 95 °C (30 s), 55 °C (30 s), 72 °C (30 s), then a final extension of 72 °C (5 min). Indexed libraries were newly purified using KAPA Pure beads and quantified with the Qubit™ 4 1× dsDNA HS Assay Kit (Thermo Fisher Scientific, Milan, Italy). The library pool was then diluted and denatured according to the Illumina MiSeq library preparation guide. The amplicon library (8 pM) was spiked with 20% denatured and diluted PhiX Control v3 (Illumina). The sequencing run was conducted on the Illumina MiSeq platform at the Department of Biology, University of Florence (Italy), using the MiSeq Reagent Kit v3 (600 cycle) with 300 bp paired-end reads. Raw sequencing data were uploaded to the European Nucleotide Archive (ENA) under the accession number PRJEB59440.

### 2.7. Clustering of 16S rRNA Amplicons

Amplicon sequences were clustered into “Amplicon Sequence Variants” (ASV) by using the DADA2 pipeline version 1.14.1 [30]. PCR primers were removed with cutadapt (version 1.15) in paired-end mode: if a primer was not found, the sequence was discarded together with its mate to reduce possible contamination [31]. Reads with more than 2 expected errors were removed using the “filterAndTrim” function of DADA2. Raw sequences were denoised with the “dada” function after error rate correction and modeling with the “learnErrors” function (both functions available in the DADA2 package). Resulting sequences were merged, discarding those with any mismatches and/or an overlap length shorter than 20bp (“mergePairs” function). Chimeric sequences were removed using the “removeBimeraDenovo” function, whereas taxonomical classification was performed using DECIPHER package [32] version 2.14.0 against the latest version of the pre-formatted Silva small-subunit reference database [33] (SSU version 138 available at http://www2.decipher.codes/Downloads.html (accessed on 19 July 2022)).

### 2.8. Metataxonomic and Diversity Analyses

The subsequent metataxonomic analyses were performed by using the R environment (version 4.1.1), with the packages Phyloseq [34], Microbiome [35], and MicrobiomeSeq [36]. Beta and alpha diversity analysis were performed on the rarified ASV table of 12,844 reads per sample. AVSs with a relative abundance >1% were considered as dominant. Richness and Shannon indices were calculated to evaluate the alpha diversity of the bacterial communities in the two areas, and beta diversity was tested by permutational analysis of variance and non-metric multidimensional scale analysis (NMDS) [37]. Furthermore, a Linear Discriminant Analysis Effect Size (LEfSe) was performed to determine the taxa most likely to explain the differences between the bacterial communities in the two areas [37].

## 3. Results

### 3.1. Cultivation and Identification of Bacteria

The viable titer of bacteria was up to 2.7 × 104 CFU/g in the propolis samples of the Arroyo area and 1.2 × 10^7^ CFU/g in the Yauco area (Table 2). The viable bacterial titer of heat-treated suspensions was at least two (hives #2 and #8) and up to four (Florencia) orders of magnitude lower than the viable titer of the corresponding untreated suspensions.

A total of 16 bacterial strains corresponding to 16 morphotypes were isolated from the mother plates of the cultivated propolis samples. The 16S rRNA was amplified from each bacterial strain. The analysis of the sequenced 16S rRNA amplicons showed the presence of 11 genera, with *Bacillus* as the most represented, with three possible species: *B. cereus* (FLORENCIA F), *Bacillus* sp. (LUCIA L), and *B. thuringensis* (LUCIA A) (Table 3).

### 3.2. Antimicrobial Activity of Propolis Extracts and Bacterial Isolates

The results of the antimicrobial activity of the propolis extracts are shown in Appendix A. Overall, all the propolis alcoholic extracts were able to inhibit the tester strains, and alcoholic propolis extracts from the Yauco area showed greater antibacterial activity against most of the testers, both Gram-positive and Gram-negative strains. The #4 aqueous extract did not show any effect against the tester strains.

Twelve bacteria isolated from propolis were tested for the production of antimicrobial compounds against six bacterial tester strains. In the colony picking test, no propolis bacteria were able to inhibit the growth of the tester strains; for this reason, a cross-streaking test was performed. The results, summarized in Table 4, showed that after 4 days of incubation at 30 °C, all the bacteria isolated from propolis were able to completely inhibit the growth of *E. coli*, while *S. marcescens* and *K. rizophyla* were partially or completely inhibited by all the bacteria except #13A. No bacteria inhibited *K. pneumonia* and *B. cereus*, while #13D, LUCIA A, and #2C were able to inhibit the growth of S. aureus. After 10 days of incubation at 30 °C, propolis bacteria showed stronger inhibition against all the tester strains. All of them were able to inhibit *E. coli* completely, and *S. marcescens* and *K. rizophyla* partially or completely. *K. pneumoniae* was partially inhibited only by FLORENCIA C and FLORENCIA F. *B cereus* and *S. aureus* were partially or completely inhibited by all propolis strains except #13A, #13F, LUCIA H, and LUCIA L.

### 3.3. Diversity and Metataxonomic Analyses of the Bacterial Communities

The bacterial community richness in the two areas was not statistically significant, but the bacterial community of Arroyo showed a Shannon index higher than that of the Yauco area (Figure 2A). However, the analysis of variance (ANOVA) showed no statistically significant differences (*p*-value > 0.05) between the two areas in terms of alpha diversity. Communities’ dissimilarity was evaluated by non-metric multidimensional scale analysis (NMDS) (Figure 2B). Results of the PERMANOVA showed a statistically significant pattern among the samples (*p* value < 0.05): samples from the same area were slightly clustered, with a stress value = 0.09, suggesting minor but statistically significant dissimilarity between the bacterial communities of the two areas.

Overall, the microbial community was composed of 7116 taxa. In propolis samples from the Arroyo area, the bacterial community was dominated by Proteobacteria (58.04%), followed by Actinobacteriota (14.66%), Cyanobacteria (11.30%), Firmicutes (6.16%), Bacteroidota (6.20%), and Acidobacteriota (1.09%) phyla (Figure 3). The Proteobacteria phylum shows the highest diversity, being composed of seven families: (Burkholderiaceae (12.47%), Enterobacteriaceae (11.08%), Sphingomonadaceae (4.90%), Beijerinckiaceae (1.46%), Erwiniaceae (4.96%), Acetobacteriaceae (1.52%), Rhizobiaceae (3.15%) and six genera (*Burkholderia-Caballeronia-Paraburkolderia* (12.19% of the total community), *Pseudoxanthomonas* (1.19%), *Pseudomonas* (1.92%), *Stenotrophomonas* (1.32%), *Pantoea* (2.06%), *Sphingobacterium* (1.80%)). Within the Actinobacteriota phylum, the most abundant genera are *Streptomyces* (1.32%) and *Corynebacterium* (1.23%) (Figure 4). In general, there is higher diversity at the genus level in the Arroyo area than in the Yauco area, with the exclusive presence of the *Pseudomonas*, *Stenotrophomonas*, and *Pantoea* genera (Figure 4). Proteobacteria is the most represented phylum even in propolis samples from the Yauco area (75.34%), followed by the Firmicutes (7.46%), Actinobacteriota (7.31%), Bacteroidota (2.31%), and Patescibacteria (1.01%) phyla (Figure 3). The phylum with the highest diversity is again Proteobacteria, constituted by four families, Burkholderiaceae (28.71%), Enterobacteriaceae (16.24%), Sphingomonadaceae (3.62%), and Beijerinckiaceae (2.68%), and three genera, *Burkholderia-Caballeronia-Paraburkholderia* (34.12% of the total community), *Arsenophonus* (13.98%), and *Methylobacterium-Methylorubrum* (2.02%). The Actinobacteria phylum is represented by the *Pseudonocardia* genus (1.56%) (Figure 4).

It is worth noting that approximately 30% of the community in the Yauco area and approximately 45% of the community in the Arroyo area were not resolved at the genus level by the 16S rRNA sequencing (NA, Figure 4).

Linear Discriminant Analysis Effect Size (LEfSe) found some indicator taxa associated with Yauco (dry forest) and Arroyo (moist forest) areas (Appendix A). The genus *Pseudonocardia* and the family Rhodocyclaceae with *Dechloromonas* were more abundant in the dry forest. Flavobacteriales, Sphingobacteriales, Xanthomonadales, and Verrucomicrobiales were more abundant in the moist forest. In addition, Rhodocyclaceae, Brevibacteriaceae, Flavobacteriaceae, Weeksellaceae, Sphingobacteriaceae, Erwiniaceae, Pseudomonadaceae, and Xanthomonadaceae were the families associated with the moist forest, while, at a finer taxonomic level, *Pseudonocardia*, *Dechloromonas*, *Brevibacterium*, *Chryseobacterium*, *Rhizobium*, *Sphingobium*, and *Pseudoxanthomonas* were the genera associated with the dry forest.

## 4. Discussion

In this work, the diversity and the antimicrobial potential of the bacterial community associated with tropical propolis from Puerto Rico were investigated for the first time by both cultivation and 16S rRNA metataxonomic approaches.

Propolis was sampled in two different geographic areas with arid (Yauco area) and moist environments (Arroyo area) in Puerto Rico. For bacterial cultivation, standard laboratory conditions similar to those previously adopted for Mediterranean propolis were followed [4]. The viable titer of bacteria determined for samples of six hives was variable, up to 2.7 × 10^4^ CFU/g of propolis for the Arroyo area and 1.2 × 10^7^ CFU/g of propolis for the Yauco area, slightly higher than those found in Mediterranean propolis [4]. The variability of the viable titer may be partly due to propolis being a sticky material that is difficult to dissolve in a solution. The heat resistance test showed the presence of a large proportion (hives #2 and #8) up to the totality (Florencia) of cells in a heat-sensitive vegetative state. According to Casalone et al. (2020) [4], we confirmed the presence of viable bacteria in different physiological states, some of which are cultivable, also compatible with the ability to perform metabolic activity inside propolis.

The identification of the isolated bacteria revealed the presence of 11 genera. Some of the identified bacteria were already found in other beehive components (such as pollen, honey, fresh stored nectar, bee bread) or insects, such as the ubiquitous species *Stenotrophomonas maltophilia*, found as an endosymbiont in bees; *Enterobacter cloacae*, very common in insects and in the larval gut; and the *Bacillus* genus, which is one of the components of the core microbiome of the bee’s gut and also plays an important role in defense and the conservation of honey [38]. The other bacteria identified were found in different habitats, compatible with those visited by bees during the foraging process: the *Curtobacterium* genus (#13F and #13A) is common in soil and also as epiphytic bacteria in several plants [38], while the *Oceanobacillus* genus (LUCIA C and #13D), *Luteimonas* (Florencia A), *Alkalihalobacillus* genus (FLORENCIA B), and *Lysinibacillus odyssey* (LUCIA H) live typically in salty and marine environments, such as hypersaline lakes, coastal regions, and soil [39,40]. *S. saprophyticus* (#2C) and *S. kloosii* (Florencia E) are very common in the skin and in the gastrointestinal flora of various animals [41], while the *Rhodoccus* genus (#13G) has been isolated from a large variety of sources, including soils, rocks, marine sediments, and animal dung, and in the gut of *Apis florea*, a wild and open nested bee [42]. Lastly, it is interesting to note the presence of *Citricoccus parietis* (FLORENCIA C), first isolated in 2010 from the mold-colonized wall of a house in Germany [43], but no more information about this species is available in the literature.

In studies focused on evaluating the antimicrobial activity of propolis, alcoholic extracts are usually prepared starting from tens of grams of samples [4,44]. Given the limited amounts of propolis samples (approximately 2 g per sample) and the multiple experimental uses, we prepared the propolis extracts with 100 milligrams of propolis. Despite this very low amount, we could test the antimicrobial activity of the propolis extracts by using an antibiogram-like test and spotting a small volume of extract (5 µL). All the propolis alcoholic extracts showed antimicrobial activity against the tester strains, better as undiluted solutions, while the water extract of sample #4 gave no inhibition of bacterial growth. Overall, the inhibitory effect was slightly higher against Gram-positive bacteria, in line with the results reported in the literature for propolis from tropical areas, especially Brazilian propolis, showing that alcoholic extracts inhibit different tester strains with a higher inhibitory effect on Gram-positive bacteria [2,44]. Seidel et al. (2008) [45] demonstrated that propolis from tropical areas shows higher antimicrobial activity than propolis from other areas, and this could be explained by the higher content of flavonoids and phenolic acids, but also by the environmental conditions. The constantly moist and warm environment typical in tropical regions favors the development of rich vegetation, and the abundance of water surfaces favors microbial endophytic and epiphytic growth.

These plant-associated microorganisms (mainly fungi and bacteria) can produce bioactive molecules as secondary metabolites, involved in host–microorganism communication and the competition between microbial species [45], which are collected by worker bees during the foraging process and the collection of the resin.

While there are several studies about the antimicrobial activity of tropical propolis, there is very little information available about the antimicrobial activity of the associated microorganisms. The results of the present study demonstrate that all the isolated bacteria showed antimicrobial activity, being able to inhibit at least the tester strain *E. coli* in the cross-streaking test, in accordance with what was previously found for the bacterial strains isolated from honey [46] and from Mediterranean propolis [4]. In the colony picking test, where propolis strains grew for 24 h, no inhibition of tester strains was observed. On the other hand, in the cross-streaking test, we observed growth inhibition increasing with the incubation time of the propolis strains, with partial or light inhibition after 4 days of incubation and a higher level of inhibition after 10 days of incubation. This is in accordance with what Casalone et al. (2020) [4] found and what is known about antimicrobials as secondary metabolites, the production of which depends on the growth phase and the physiological state of the bacterial cells [47].

Overall, the inhibitory effect was stronger against Gram-positive tester strains, and the most resistant tester strain was *K. pneumoniae*, in line with what was previously found in Casalone et al. (2020) [4]. FLORENCIA C (*Citricoccus parietis*), LUCIA A (*Bacillus thuringensis*), and FLORENCIA F (*Bacillus cereus*) showed the strongest antimicrobial activity, inhibiting completely or partially all six tester strains. These strains, with high antimicrobial potential, come from hives of the Arroyo area, which is in the moist forest, where the lush vegetation and abundant rain favor microbial proliferation, so that competition between microbial species is fierce and the production of antimicrobial compounds could be a winning strategy to survive. While, in the literature, there is no information about *C. parietis*, *Bacillus cereus*, as with most of the *Bacillus spp*., is recognized as being an antimicrobial producer. Several studies demonstrate how it can inhibit different microorganisms, both fungi, such as *Fusarium solani* [48], and Gram-positive and Gram-negative bacterial strains [49,50]. Furthermore, *B. cereus* and other species belonging to *Bacillus* showed the ability to inhibit the growth of *Paenibacillus larvae*, which causes American foulbrood, one of the most fatal diseases for the larvae of honeybees [51], highlighting the potential role of these bacteria as biological control agents. These results support the hypothesis that the propolis microbiota could play a role in propolis’ properties, enhancing its intrinsic antimicrobial activity and protecting the hive against pathogenic microorganisms [4].

To further investigate the structure and the diversity of the bacterial community, a 16S rRNA metataxonomic analysis was performed on 10 propolis samples, five from each area. While there was no statistical difference in the number of ASVs in the propolis from the two areas, NMDS analysis showed that there was a statistically significant clustering between samples from Yauco and Arroyo (*p* value < 0.05), indicating dissimilarity in the microbial communities hosted by propolis from the two areas. Hence, even if the number of taxa associated with propolis is quite similar, their compositions in the two communities are different, probably because of the different climate conditions, which imply different dominant flora and, consequently, different dominant microbes. The bacterial communities of both Yauco and Arroyo samples are dominated by Proteobacteria, similarly to what was found for Mexican and Mediterranean samples [4,21]. While the dominant phyla of Proteobacteria, Actinobacteriota, and Firmicutes were already found [4,21,23], surprisingly, Cyanobacteria have never been found in propolis, although they have been found in the bee’s gut [52].

Within the Proteobacteria associated with the Yauco propolis, Burkholderiaceae and Enterobacteriaceae are the dominant families, while the Proteobacteria community of the Arroyo propolis is composed mainly of the Enterobacteriaceae, Erwiniaceae, and Sphingomonadaceae families and shows higher diversity. Enterobacteriaceae and Sphingomonadaceae are associated with the guts of larvae, adult honeybees, and other bee products, such as pollen, honey, and beebread [38]. In fact, genera of Enterobacteriaceae are often detected as epiphytic of numerous plants, especially of flowers, and honeybees may come into contact with them during the pollen and nectar collection process. In particular, *Pantoea* genus, present in samples from the Arroyo area, comprises species already detected in pollen loads, honey sacs, and freshly stored nectar in honeybee colonies [53], and some isolates are able to produce antimicrobials [54]. Genera of Erwiniaceae have been found to be associated with corbicolar pollen [17] and also with Mediterranean and Turkish propolis [4,23].

At the genus level, the difference between the two communities is appreciable. In Yauco’s samples, two genera are dominant: *Burkoldheria-Caballeronia-Paraburkoldheria* and *Arsenophonus* (belonging to the Morganellaceae family). *Burkoldheria* belongs to the larger and taxonomical complex group of *Burkoldheria sensu lato*, which comprises strains from a very wide range of environmental habitats, such as soil, water, plants, and fungi [55], while the *Arsenophonus* genus comprises endosymbiotic species of several insects, and it has been detected in the gut microbiota, body surface, and hemolymph of worker bees. It seems to be more abundant in worker bees belonging to colonies with Colony Collapse Disease symptoms [56], a phenomenon that occurs when worker bees leave the hive, leaving the queen, the food stored, and larvae behind [57]. It is worth noting the presence of *Paenibacillus*, belonging to the Firmicutes phylum, which can constitute biotic stress for the bee colonies. Indeed, among the *Paenibacillus* genus, there are species that could be lethal pathogens for bees, such as *Paenibacillus larvae*, the etiological agent of the American foulbrood [51]. Actinobacteriota phylum is mostly represented by the *Pseudonocardia* genus, which produces antimicrobials that can help against pathogen species, and it often engages symbiosis or mutualistic associations with insects. One of the most well-known cases is the ancient symbiosis between the fungus-growing ants and *Pseudonocardia*, which produces antibiotics that can defend the mutualistic fungus from parasitic infections [58].

In Arroyo’s samples, the bacterial community at the genus level shows higher diversity than in the Yauco area, comprising genera already detected in honeybees or hive environments, such as *Corynebacterium* (Actinobacteriota) [16], detected also in Mexican propolis [21], and *Pseudomonas* (Proteobacteria) [21]. The main representative of the Actinobacteriota phylum, together with *Corynebacterium*, is the *Streptomyces* genus, commonly isolated from foraging bees, especially in stored pollen, and known to be an antimicrobial producer with great inhibitory activity against *Paenibacillus larvae* [58]. Moreover, it is interesting to note the presence of the *Mycobacterium* genus, closely related to the *Corynebacterium* genus [59] and found also in Turkish propolis [23], which comprises pathogen species for humans. It is common in soil, bogs, surface water, groundwater, and seawater [60]; hence, it may be vectored by different insect species.

Although there are still too few studies, the differences found in the bacterial communities associated with other propolis samples [4,21,23] could be partly due to the different local flora and plant sources, as well as to the different honeybee subspecies producers; likewise, the gut microbiota can differ among different honeybee species and subspecies [61]. However, further studies are needed to assess whether propolis samples produced in different environmental conditions could share a core microbiota.

It is worth noting that a large part of the bacterial community of both areas, a little less than 50% in Arroyo, was not resolved by 16S rRNA sequencing at the genus level, and many bacteria remain uncharacterized, meaning that propolis could constitute a possible natural source of new microbial strains and, considering the results of the sensitivity tests, new antimicrobial compounds useful for the control of dangerous microorganisms for honeybees, plants, and humans. Considering the decline in the worldwide bee populations [62], the search for bacteria of the beehive able to produce antimicrobial compounds against bee pathogens is of great interest. In this view, the isolation of bacterial strains inhabiting propolis and contributing to the antimicrobial properties of propolis could be useful to develop novel control agents to further enhance the protection of the whole hive. Cultivation conditions mimicking the hive and propolis environment and/or suggested by the metataxonomic analysis could help to isolate new bacterial strains inhabiting propolis.

## 5. Conclusions

With this study, the presence of a viable bacterial community with an appreciable diversity in tropical propolis was demonstrated, confirming what was found by Grubbs et al. (2015) for North American propolis [20] and by Casalone et al. (2020) [4] for Mediterranean propolis. The metataxonomic analysis revealed greater bacterial diversity and a statistically significant difference in the bacterial community compositions of the two study areas, probably due to the different climate conditions and the consequently different dominant flora. The isolated bacteria and the 16S rRNA targeted metagenomics showed the presence of bacteria already detected in other hive components, such as pollen, beebread, and the bee’s gut, and in different foraging environments. The antimicrobial activity of the bacteria isolated from propolis was ascertained, confirming what was previously found for bacteria isolated from Mediterranean propolis [4] and from honey [46]. Moreover, the antimicrobial activity of alcoholic propolis extracts against bacterial tester strains was demonstrated by preparing the extracts with a small amount of propolis. These results, obtained for propolis produced by the gentle Africanized honeybee, support the hypothesis that the propolis microbiota could play a role in propolis’ properties, enhancing its intrinsic antimicrobial activity and protecting the hive against pathogenic microorganisms.

## Figures and Tables

**Figure 1 microorganisms-11-01130-f001:**
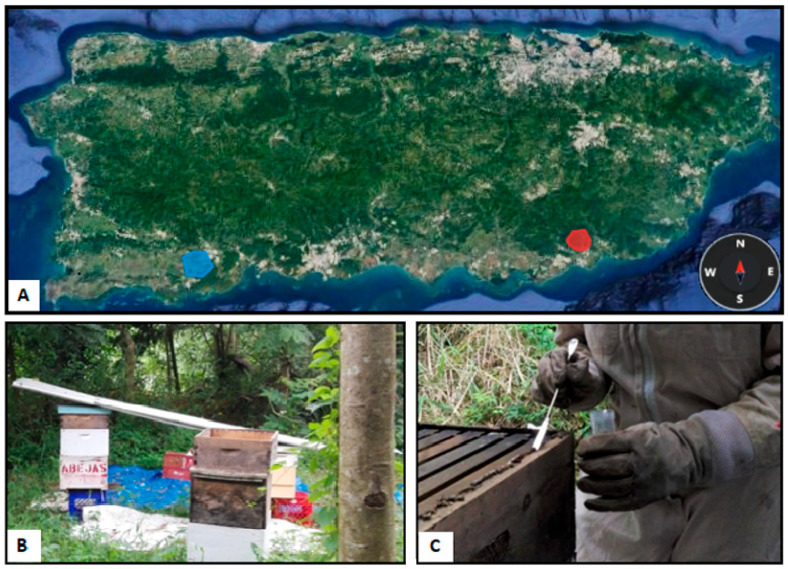
Propolis sampling sites. (**A**) Map of Puerto Rico Island showing the locations of the apiaries in Yauco (colored blue) and Arroyo (colored red) areas. (**B**) Detail of some hives of the Arroyo area. (**C**) Detail of propolis sampling from the edge of the hive structure.

**Figure 2 microorganisms-11-01130-f002:**
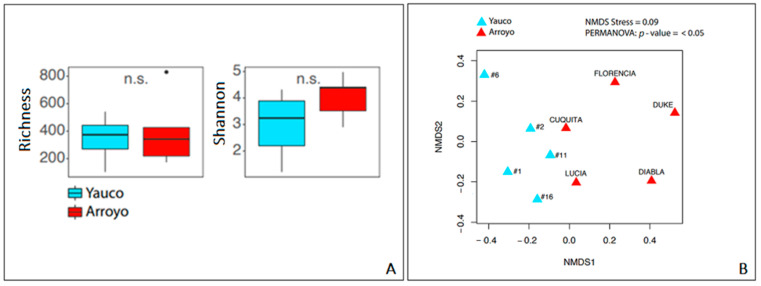
(**A**) Alpha-diversity-index-based analysis: richness and Shannon index. (**B**) NMDS ordination plots showing the dispersion of samples by their dissimilarity.

**Figure 3 microorganisms-11-01130-f003:**
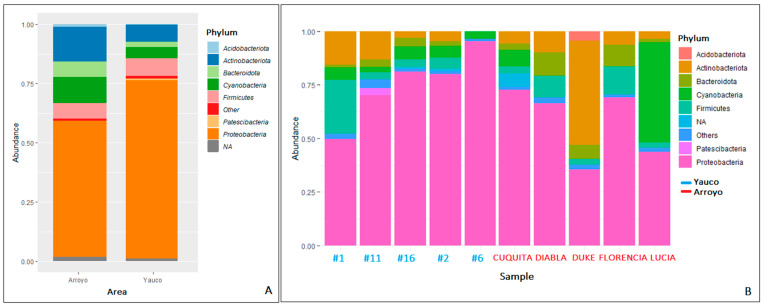
Microbial composition analysis at phylum level averaged by area (**A**) and sample (**B**). Each bar represents the relative abundance of a bacterial phylum. Phyla with an abundance <1% were grouped under the label “Others”. NA = not available.

**Figure 4 microorganisms-11-01130-f004:**
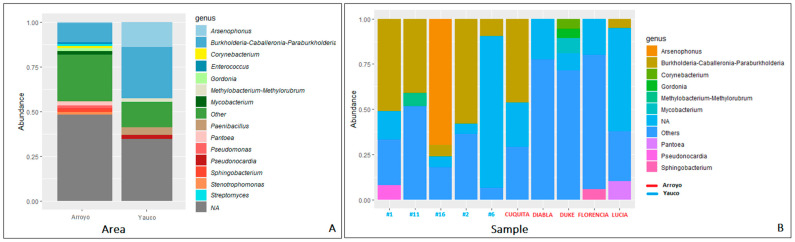
Microbial composition analysis at genus level averaged by area (**A**) and sample (**B**). Each bar represents the relative abundance of a bacterial genus. Genera with an abundance <1% were grouped under the label “Others”. NA = not available.

**Table 1 microorganisms-11-01130-t001:** List of propolis samples and their processing for this work. ✓ indicates the analyses done.

Area	Sample (Hive)	Metataxonomic	Cultivation	Ethanolic Extracts
Yauco	#1	✓	✓	✓
	#2	✓	✓	✓
	#4			✓
	#6	✓		
	#8		✓	✓
	#11	✓		✓
	#13		✓	✓
	#16	✓		
Arroyo	AKIRA			✓
	CUQUITA	✓		✓
	DIABLA	✓		✓
	DUKE	✓		✓
	FLORENCIA	✓	✓	✓
	LUCIA	✓	✓	✓

**Table 2 microorganisms-11-01130-t002:** Average viable titer in CFU/g of untreated and heat-treated suspensions of bacteria from propolis samples. nm = not made; nd = not detectable, no colonies grew.

Area	Hive	Untreated (CFU/g ± Sd)	Heat-Treated (CFU/g ± Sd)
Arroyo	Lucia	1.0 × 10^3^ ± 1.5 × 10^2^	nm
	Florencia	2.7 × 10^4^ ± 6.1 × 10^3^	nd
Yauco	#1	1.2 × 10^7^ ± 3.1 × 10^6^	nm
	#2	4.0 × 10^6^ ± 9.8 × 10^5^	2.5 × 10^4^ ± 1.2 × 10^4^
	#8	1.0 × 10^6^ ± 2.6 × 10^5^	7.5 × 10^3^ ± 1.1 × 10^4^
	#13	2.8 × 10^4^ ± 2.0 × 10^3^	nm

**Table 3 microorganisms-11-01130-t003:** Identification of bacteria isolated from propolis samples by 16S rRNA gene analysis.

Area	Strain/ Morphotype *	Organisms with the Most Similar 16S rRNA Sequences	Identity (%)
Yauco	#1C	*Stenotrophomonas maltophilia*	98.59
	#2C	*Staphyloccoccus saprophyticus*	98.30
	#8C	*Enterobacter ludwigii*	98.58
	#13A	*Curtobacterium luteum*	98.43
	#13D	*Oceanobacillus profundus*	98.24
	#13F	*Curtobacterium oceanosedimentum*	98.00
	#13G	*Rhodococcus baikonurensis*	99.02
Arroyo	FLORENCIA A	*Luteimonas* sp.	98.09
	FLORENCIA B	*Alkalihalobacillus xiaoxiensis*	98.58
	FLORENCIA C	*Citricoccus parietis*	98.06
	FLORENCIA E	*Staphylococcus kloosii*	99.53
	FLORENCIA F	*Bacillus cereus*	99.21
	LUCIA A	*Bacillus thuringensis*	97.21
	LUCIA C	*Oceanobacillus sojae*	97.48
	LUCIA H	*Lysinibacillus odysseyi*	98.82
	LUCIA L	*Bacillus toyonensis*	98.15

* The name of the morphotype corresponds to the hive name followed by a capital letter; the same letter does not identify the same morphotype in different hives.

**Table 4 microorganisms-11-01130-t004:** Results of the cross-streaking test after 4 and 10 days of incubation at 30 °C of the bacteria isolated from propolis. Legend: − = no inhibition (growth); + = complete inhibition (no growth); +/− = partial inhibition (partial growth).

Bacterial Strains from Propolis	Tester Strains											
	*E. coli*		*S. marcescens*		*K. rizophyla*		*K. pneumoniae*		*B. cereus*		*S. aureus*	
	4	10	4	10	4	10	4	10	4	10	4	10
#2C	+	+	+/−	+/−	+/−	+	−	−	−	+	+/−	+
#13A	+	+	−	+/−	−	−	−	−	−	−	−	−
#13D	+	+	+/−	+/−	+	+	−	−	−	+/−	+/−	+
#13F	+	+	+/−	+/−	−	−	−	−	−	−	−	−
#13G	+	+	+/−	+	+/−	+/−	−	−	−	+	−	+
FLORENCIA B	+	+	+/−	+	+/−	+/−	−	−	−	+/−	−	+/−
FLORENCIA C	+	+	+	+	+	+	−	+/−	−	+	−	+
FLORENCIA F	+	+	+/−	+	+	+	−	+/−	−	+	−	+
LUCIA A	+	+	+/−	+	+/−	+	−	−	−	+	+	+
LUCIA C	+	+	+/−	+	+/−	−	−	−	−	+/−	−	+
LUCIA H	+	+	+/−	+	+/−	+/−	−	−	−	−	−	−
LUCIA L	+	+	+/−	+	+	+	−	−	−	−	−	−

## Data Availability

Not applicable.

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
