# Peer review of "Characterization of the Bacterial Communities Inhabiting Tropical Propolis of Puerto Rico"

_microorganisms, 2023, doi:10.3390/microorganisms11051130_

Round 1

Reviewer 1 Report

The paper entitled Characterization of the bacterial communities inhabiting tropical propolis of Puerto Rico describes the diversity of bacterial population and the antimicrobial potential of the propolis from Puerto Rico. These type of investigations are needed, because there is a lack of scientific studies describing the bacterial populations from bee products in general, and propolis in particular.  The paper is well written, methods are thoroughly described, results also and discussions are made comparing own results with existing literature. Excepting few minor errors (superscript numbers in the text, italic writing in some cases), a clarification (what means LEfse, page 10, line 387), I am recommending the Editorial office to take into consideration the publication of the study

Author Response

Dear Reviewer,

We greatly appreciate your careful reading of our text and your thoughtful comments which have been very helpful in improving the manuscript. We are also grateful for the overall positive appreciation of the manuscript. We trust that all your comments have been addressed accordingly in the revised manuscript. We list here below a point-by-point reply. Throughout, your comments are in blue and our response in black font. The line numbers refer to the revised manuscript. The corresponding modified parts of the text are marked up using the “Track Changes” function and highlighted in grey in the revised manuscript.

Excepting few minor errors (superscript numbers in the text, italic writing in some cases), a clarification (what means LEfse, page 10, line 387), I am recommending the Editorial office to take into consideration the publication of the study.

Superscript numbers and italic writing have been checked and corrected.

The meaning of LEfSe is reported in 2.8 (lines 312-313), we added its full name again on line 393.

Reviewer 2 Report

Comments for authors

1.      In line 18: The authors are talking about studies already done, so the term should be in the past tense "showed".

2.      In line 24: please change the terms "we, I, our ........" to the passive form. For example, the sentence could be "The present study revealed that ........."

3.      Authors should add a brief of the results in the abstract. The abstract needs more improvement in general.

4.      The introduction is so long, please shorten and focus on the most relevant topics.

5.      In lines 48-50: This part is not necessary because the study does not include nutritional part or biological activity other than antimicrobial potential.

6.      In lines 122-123, How to explain antimicrobial potential of the isolated bacteria? In lines 128-130, this sentence of the method will not add any necessary information to the objective of the study, please remove.

7.      In Table 1: The authors did not mention the steps of ethanolic extraction of the propolis samples.

8.      In Figure 1: These (A) and (B) pictures should be deleted too because they did not add something new to the study.

9.      Authors should follow the instructions of authors, for e.g., figure captions, numbering…etc.

10.  Some of the inserted references need to be updated.

11.  Authors could benefit from the following reference in the introduction: Yosri, N., El-Wahed, A., Aida, A., Ghonaim, R., Khattab, O. M., Sabry, A., ... & El-Seedi, H. R. 2021: Anti-Viral and Immunomodulatory Properties of Propolis: Chemical Diversity, Pharmacological Properties, Preclinical and Clinical Applications, and In Silico Potential against SARS-CoV-2. Foods, 10(8), 1776.

Author Response

Dear Reviewer,

We greatly appreciate your careful reading of our text and your thoughtful comments which have been very helpful in improving the manuscript. We trust that all your comments have been addressed accordingly in the revised manuscript. We list here below a point-by-point reply. Throughout, your comments are in blue and our response in black font. The line numbers refer to the revised manuscript. The corresponding modified parts of the text are marked up using the “Track Changes” function and highlighted in yellow in the revised manuscript.

  1. In line 18: The authors are talking about studies already done, so the term should be in the past tense "showed".
  2. In line 24: please change the terms "we, I, our ........" to the passive form. For example, the sentence could be "The present study revealed that ........."
  3. Authors should add a brief of the results in the abstract. The abstract needs more improvement in general.

Thank you for your suggestions. The abstract has been modified in accordance with the requirements of points 1, 2 and 3. The main results are reported on lines 23-28.

  1. The introduction is so long, please shorten and focus on the most relevant topics.
  2. In lines 48-50: This part is not necessary because the study does not include nutritional part or biological activity other than antimicrobial potential.

Thank you for your suggestions. The introduction has been modified in accordance with the requirements of points 4 and 5. We deleted lines 46-51 (including 48-50) and lines 53-56 of the previous manuscript because they concerned chemical composition and biological activities other than antimicrobial potential that are not relevant topics to our work.

  1. In lines 122-123, How to explain antimicrobial potential of the isolated bacteria?

In a previous work (Casalone et al., 2020) we isolated bacteria from Italian propolis and found that they showed antimicrobial activity against Gram-negative and Gram-positive bacteria and entomopathogenic fungi, with different inhibition spectra. Our hypothesis is that propolis microbiota could contribute to propolis's overall antimicrobial properties.

In lines 128-130, this sentence of the method will not add any necessary information to the objective of the study, please remove.

The sentence is: ‘The Arroyo area is in the Moist Forest, where rains are abundant, and evergreen vegetation with very tall trees is present. Yauco area is in the Dry Forest, where rains are scarce, the vegetation covers the ground, and fire is very common.’

We agree that this sentence does not add any necessary information to the objective of the study. On the other hand, it adds information about description of the study areas; for this reason, we preferred to keep it in the section 2.1. Study areas and hives description and sampling.

  1. In Table 1: The authors did not mention the steps of ethanolic extraction of the propolis samples.

The steps of ethanolic extraction of the propolis samples are described in section

2.4.

  1. In Figure 1: These (A) and (B) pictures should be deleted too because they did not add something new to the study.

We deleted the (B) picture. Even if the picture (A) doesn’t add something new, we preferred to keep it because it can be helpful to the reader to localize the two geographic areas were the propolis was sampled. We hope you agree with us.

  1. Authors should follow the instructions of authors, for e.g., figure captions, numbering…etc.

Thank you for this comment. We revised the manuscript according to the instructions for authors of the Journal and the template we received from the Editorial Office.

  1. Some of the inserted references need to be updated.

Thank you for your comment. We noticed some mistakes in the references, maybe due to the incorrect use of the software Mendeley. We sincerely apologize for that. We proceeded to update all the references.

  1. Authors could benefit from the following reference in the introduction: Yosri, N., El-Wahed, A., Aida, A., Ghonaim, R., Khattab, O. M., Sabry, A., ... & El-Seedi, H. R. 2021: Anti-Viral and Immunomodulatory Properties of Propolis: Chemical Diversity, Pharmacological Properties, Preclinical and Clinical Applications, and In Silico Potential against SARS-CoV-2. Foods, 10(8), 1776.

Thank you for the suggestion. We added the reference in the Introduction section on line 43 (re. 3).

Reviewer 3 Report

Dear Editor,

The authors of this manuscript investigate the bacterial communities of propolis in two different geographic areas of Puerto Rico (PR, USA). I have some suggestions and recommendations to improve quality of the paper.

Line 22. Metataxonomic analysis – Please, replace this with matabarcoding analysis.

Line 59. Any citations to support these statements?

Line 61. Please, see the note above.

Line 64. In pointed out reference [7], I see research regarding bacterial microbiome of propolis, but not any active against different bacterial species!!!

Line 65. This reference is associated with the honey bee gut microbiome, but not propolis bacterial community!!!

Line 106. Again, this reference is completely wrong – in this research, the authors studied atibacterial compound- from marine bacterial species, not fungi.

Line 111. several European Apis mellifera subspecies – please, mention some of them.

Line 119-121. Please, move this sentence in Materials and Methods section.

Line 179. Stereomicroscope – please, add a manufactural for that.

Line 180. Please, draw a Table included alphabetical letters for each hives and regions. This would make things much easier for the reader.

Line 185. Zymo Research – city, country?

Line 187-188. Which variable regions of 16S rRNA gene covered these primers? Please, add additional information.

Line 210. at 37°C for a night – may be overnight.

Line 295. Please, add this reference in the end of the sentence:

Segata, N.; Izard, J.; Waldron, L.; Gevers, D.; Miropolsky, L.; Garrett, W.S.; Huttenhower, C. Metagenomic biomarker discoveryand explanation. Genome Biol. 2011, 12, R60.

Line 307. B. cereus – please, give a species name in italic.

From line 337 and line 344 is obviously that the propolis extracts inhibited growth both Gram-positive and Gram-negative strains, but in the Abstract section the author’s state more inhibitory effect on Gram-positives bacterial strains. Please, correct that.

Line 361. How the authors interpret the different pattern of clustering in samples from the same apiary?

Line 364. From fig. 2B I can not see statistically significant dissimilarity between the bacterial communities of the two areas. Rather, there is a similarity between the bacterial diversity in some hives from the two regions.

Line 376. From Fig. 4. I have noticed that the genera Pseudomonas and Pseudonocardia prevailed in Yauco area.

Line 440. beehive components - what the authors mean by mentioning that?

Line 505. This citation is not related with the growth of Paenibacillus larvae. In this regard, I advise the authors to check all cited sources and make appropriate changes

The Reference section is not prepare according Journal requirement.

Author Response

Dear Reviewer,

We greatly appreciate your careful reading of our text and your thoughtful comments which have been very helpful in improving the manuscript. We trust that all your comments have been addressed accordingly in the revised manuscript. We list here below a point-by-point reply. Throughout, your comments are in blue and our response in black font. The line numbers refer to the revised manuscript. The corresponding modified parts of the text are marked up using the “Track Changes” function and highlighted in green in the revised manuscript.

Line 22. Metataxonomic analysis – Please, replace this with matabarcoding analysis.

Done.

Line 59. Any citations to support these statements?

Line 61. Please, see the note above.

We added ref. 7 on line 61, ref. 8 on line 62 and ref. 4 on line 63. We also slightly modified the sentence on line 62 from: ‘The most studied aspect is its antimicrobial activity’ in: ‘One of the most studied aspects is its antimicrobial activity’.

Line 64. In pointed out reference [7], I see research regarding bacterial microbiome of propolis, but not any active against different bacterial species!!

Line 65. This reference is associated with the honey bee gut microbiome, but not propolis bacterial community!!!

Line 106. Again, this reference is completely wrong – in this research, the authors studied atibacterial compound- from marine bacterial species, not fungi.

Thank you for your remarks on references! The mistakes in the references maybe due to the incorrect use of the software Mendeley. We sincerely apologize for that. We checked that all the citations in the text match those in the References section.

Line 111. several European Apis mellifera subspecies – please, mention some of them.

We added the required information on line 114 and added ref. 25 (Hall et al., 1990).

Line 119-121. Please, move this sentence in Materials and Methods section.

We deleted the sentence from Introduction and better specified the number of apiaries and hives in Materials and Methods section, line 130.

Line 179. Stereomicroscope – please, add a manufactural for that.

Done, see line 194.

Line 180. Please, draw a Table included alphabetical letters for each hives and regions. This would make things much easier for the reader.

The requested information is already reported in Table 1 (areas and hives/samples) and Table 3 (areas and hives/morphotypes for each hive). To make it clearer to the reader, we added a footnote in Table 3 with the explanation of the name of the morphotypes. For this reason, we preferred not to add another table.

Line 185. Zymo Research – city, country?

We added the requested information, see line 201.

Line 187-188. Which variable regions of 16S rRNA gene covered these primers? Please, add additional information.

The whole 16S rRNA gene was amplified by primers 8F and 1492R. We specified that on line 202.

Line 210. at 37°C for a night – may be overnight.

Done, see line 226.

Line 295. Please, add this reference in the end of the sentence:

Segata, N.; Izard, J.; Waldron, L.; Gevers, D.; Miropolsky, L.; Garrett, W.S.; Huttenhower, C. Metagenomic biomarker discoveryand explanation. Genome Biol. 2011, 12, R60.

Thank you for the suggestion. The reference was added on line 311.

Line 307. B. cereus – please, give a species name in italic.

Done, lines 323-324.

From line 337 and line 344 is obviously that the propolis extracts inhibited growth both Gram-positive and Gram-negative strains, but in the Abstract section the author’s state more inhibitory effect on Gram-positives bacterial strains. Please, correct that.

The abstract was corrected in accordance with this request (line 28)

Line 361. How the authors interpret the different pattern of clustering in samples from the same apiary?

Since most of the bacteria found in propolis are presumably vectored from plant resins (ref. 4 in the manuscript and the present study), one of the factors influencing the clustering pattern and variability of the propolis microbiome within the same areas could be the high plant biodiversity and, thus, the broad range of plants where the bees can potentially forage.

Line 364. From fig. 2B I can not see statistically significant dissimilarity between the bacterial communities of the two areas. Rather, there is a similarity between the bacterial diversity in some hives from the two regions.

We know the dissimilarity between propolis bacterial communities of the two areas visualized by NMDS is slight. Thus, we tested differences via PERMANOVA (permutational multivariate ANOVA) and we got a p-value < 0.05. We reported that the difference at the community level is not strong but still significant (lines 366-370). To better understand the potential differences, we used the linear discriminant analysis effect size (LEfSe) method. LEfSe detects the taxa most likely to explain differences between sites by coupling standard tests for statistical significance with additional tests encoding biological consistency and effect relevance.

Line 376. From Fig. 4. I have noticed that the genera Pseudomonas and Pseudonocardia prevailed in Yauco area.

In Fig. 4A Pseudomonas is present among the most abundant genera in the Arroyo area, while Pseudonocardia in the Yauco area. They are marked by two different shades of red, we agree that similar colors are confusing. 

Line 440. beehive components - what the authors mean by mentioning that?

We specified them ay lines 445-446.

Line 505. This citation is not related with the growth of Paenibacillus larvae. In this regard, I advise the authors to check all cited sources and make appropriate changes

The Reference section is not prepare according Journal requirement.

Please see the above response to comments to lines 64, 65, 106.

We checked all the references and modified them according to the Journal requirements.

Round 2

Reviewer 3 Report

Dear Editor,

The authors followed all my suggestions and recommendations and manuscript can be accepted now.